Development of a prognostic RiskScore model using efferocytosis-related signature genes for lung adenocarcinoma

Dai Zengmin
Jin Shaofeng
Huang Shanshan
Liu Bingyang
Shen Xingkai
Jin Yuhong jinyuhong2021@outlook.com
Department of Critical Care Medicine, Ningbo Medical Center Lihuili Hospital (The Affiliated Lihuili Hospital of Ningbo University) , Ningbo , China
Guan Fanglin
Electronic publication date: 2025 Sep 5
Publication date: 2025
Volume: 13
Electronic Location ID: e19892
Received 2025 Apr 16; Accepted 2025 Jul 21
Copyright: ©2025 Dai et al.
Copyright year: 2025
Copyright holder: Dai et al.
License: This is an open access article distributed under the terms of the Creative Commons Attribution License, which permits unrestricted use, distribution, reproduction and adaptation in any medium and for any purpose provided that it is properly attributed. For attribution, the original author(s), title, publication source (PeerJ) and either DOI or URL of the article must be cited.
License URL: https://creativecommons.org/licenses/by/4.0/

Keywords: Efferocytosis, Lung adenocarcinoma, RiskScore model, Prognostic signature, Immune infiltration, Drug sensitivity

Funding: Zhejiang Provincial Medical and Health Science and Technology Project 2021KY305 The study was supported by Zhejiang Provincial Medical and Health Science and Technology Project (2021KY305). The funders had no role in study design, data collection and analysis, decision to publish, or preparation of the manuscript.

==============================
Background

Lung adenocarcinoma (LUAD) has high incidence and mortality rates. Efferocytosis is involved in the progression of various cancers. The current work set out to develop a prognosis signature using efferocytosis-related genes (ERGs) for LUAD.

Methods

Public databases were accessed to obtain bulk and single-cell data of LUAD. Molecular subtyping of LUAD was performed using ConsensusClusterPlus, and efferocytosis-related candidate genes were screened by weighted gene co-expression network analysis (WGCNA) in combination with differential analysis. Subsequently, Least Absolute Shrinkage and Selection Operator (LASSO) regression was applied to construct a prognostic RiskScore model, followed by evaluating the relationship between the RiskScore, immune infiltration, and drug sensitivity. Single-cell transcriptomic profiling of LUAD was performed with the Seurat package to elucidate the cellular origins of the key genes. The expression and potential function of the representative genes were verified by reverse transcriptional quantitative polymerase chain reaction (qPCR) (RT-qPCR), Cell Counting Kit-8 (CCK-8), wound healing, and Transwell assays.

Results

Two molecular subtypes of LUAD with different outcomes and clinical features were identified. Candidate ERGs were mainly enriched in inflammatory and immune-related pathways. Subsequently, seven key genes (CD200R1, BTN2A2, STAP1, DNASE2B, SAMD9, SEMA7A, BIRC3) associated with efferocytosis in LUAD were identified to establish a RiskScore model, which exhibited high robustness in predicting patient prognosis. Notably, high-risk group had lower immune scores and more unfavorable prognosis than low-risk group. Moreover, eight drugs were closely linked to the RiskScore, and low-risk group was more sensitive to Doramapimod_1042. Single-cell atlas of LUAD showed that the prognostic ERGs were mainly expressed in mast cells. In vitro experiments revealed that most of the seven ERGs were overexpressed in LUAD cells, and that SEMA7A knockdown could suppress LUAD cell proliferation, migration and invasion.

Conclusions

Our results provided novel insights for the prognosis prediction and personalized treatment of LUAD.

Introduction

Lung adenocarcinoma (LUAD) is a frequent lung cancer subtype that primarily originates from bronchial mucosa epithelium (Cao et al., 2024). Due to its insidious early-stage symptoms and a lack of reliable indicators, majority of patients are diagnosed at advanced or metastatic stages (Chen et al., 2021). The clinical heterogeneity LUAD results in divergent therapeutic outcomes, presenting challenges to its effective treatment (Sun et al., 2024b; Guo et al., 2023). While surgery, chemotherapy and radiotherapy are the main conventional treatment of LUAD, immune checkpoint inhibitors (e.g., PD-1/PD-L1 antibodies) and targeted drugs against driver genes such as epidermal growth factor receptor (EGFR) and anaplastic lymphoma kinase (ALK) have also improved the treatment of LUAD (Shao et al., 2022; Wei, Yang & Yuan, 2024). Nevertheless, the prognosis of LUAD remains unsatisfactory due to tumor recurrence and drug resistance, with advanced-stage patients demonstrating a dismal 5-year overall survival (OS) rate of only 10–15% (Meng et al., 2024). Therefore, this study was designed to discover specific molecular markers to guide individualized treatment.

The process of efferocytosis prevents the accumulation of dead cells, inflammation, and subsequent cell necrosis. Specifically, efferocytosis unfolds in four sequential stages: (1) Dying cells recruit phagocytes through “find-me” signaling molecules; (2) “Eat-me” and “don’t-eat-me” signals enable phagocytes to distinguish between dead and viable cells; (3) Engulfment of cellular corpses by phagocytes; (4) Degradation of dying cells occurs within phagocytic vesicles. Each of these steps is tightly regulated to ensure precise and efficient clearance of apoptotic cells, thereby maintaining tissue homeostasis and avoiding inflammatory responses (Liu et al., 2025). Previous studies manifested that some efferocytosis-related molecules or pathways could profoundly affect the progression, metastasis, drug resistance and patient prognosis of numerous malignancies, such as bladder cancer (Zheng et al., 2024), colorectal cancer (Ma et al., 2024), and pancreatic cancer (Hou et al., 2024). Additionally, efferocytosis can also reshape the tumor microenvironment (TME) to induce immunosuppressive status, thereby affecting a variety of immune processes and facilitating tumor cells to evade from immune surveillance (Qiu et al., 2023). Accordingly, regulation of efferocytosis could serve as an indicator for assessing the efficacy of anticancer therapy on patients (Zhou et al., 2020). However, there are limited number of studies probing into the involvement of efferocytosis in LUAD development.

The present study downloaded LUAD samples from public database. Based on the efferocytosis-related genes (ERGs) acquired from previous study, the molecular subtypes of LUAD were classified to analyze their prognosis and clinical features. Then, the ERGs were screened to explore functional pathways involved. Subsequently, a RiskScore model was developed using the ERGs in LUAD and further validated. Moreover, the correlation between drug sensitivity, immune infiltration, and RiskScore was analyzed. Cell types implicated in LUAD progression were uncovered based on the single-cell atlas of LUAD. Additionally, the role of the prognostic genes was validated using LUAD cell lines with in vitro assays. Overall, our findings could improve the current understanding of the mechanisms of LUAD and contribute to the prognostic assessment and precise treatment of LUAD.

Material and methods

Data acquisition and preprocessing

The clinical information and RNA-seq data of The Cancer Genome Atlas (TCGA)-LUAD cohort were obtained from the UCSC xena database. Hereafter, we excluded the samples with survival time ≤30 days or without clinical follow-up data. Next, the Ensembl was transformed into the gene symbol. If there were multiple gene symbol expressions, the maximum value was taken. The expression matrix was transformed into transcript per million (TPM) format and log2-converted. Eventually, 490 primary LUAD samples and 58 paracancer control samples in TCGA-LUAD cohort served as a training set.

The clinical information and RNA-seq data of the GSE31210 dataset were downloaded from the Gene Expression Omnibus (GEO, https://www.ncbi.nlm.nih.gov/geo/query/acc.cgi?acc=GSE31210) database. The probe was converted into Symbol according to the annotation file after eliminating normal. Subsequently, 226 LUAD samples in GSE31210 dataset were acquired to serve as a validation set.

The scRNA-seq data of two LUAD samples in the GSE149655 dataset (https://www.ncbi.nlm.nih.gov/geo/query/acc.cgi?acc=GSE149655) containing was collected from the GEO database. Thereafter, to filter the single-cell data, each gene was required to be expressed in a minimum of three cells, with each cell expressing 200 genes at least. The decontX was employed to eliminate contaminated cells (comtamination ≤ 0.2) while retaining cells with feature number (nFeature_RNA) < 7,000, transcript counts (nCount_RNA) < 1,000,000, and mitochondrial gene expression percentage (percent.mito) < 40%. Afterwards, the data were normalized using SCTransform function, and RunPCA function was used to conduct principal component analysis (PCA), followed by eliminating batch effects between different samples with the “harmony” R package (Zhang et al., 2024a). See Fig. S1 for the results after quality control. The 167 ERGs used in the study were obtained from a previous study (Xie et al., 2024) and all the genes were shown in Table S1.

Identification of molecular subtypes

Univariate Cox regression analysis was conducted to screen prognosis-associated ERGs (p < 0.05) in TCGA-LUAD cohort. Thereafter, based on these ERGs, the molecular subtypes of LUAD were classified using the “ConsensusClusterPlus” R package to perform consensus clustering analysis (Jia et al., 2024), with “km” algorithm and “1-Spearman correlation” as the distance metrics. We conducted 500 bootstraps, with each bootstrap encompassing 80% of the training set patients. The cluster number (k) was set from 2 to 10 to determine the optimal classification. Furthermore, the rationality of subtype classification was evaluated by PCA. The OS rate of different molecular subtypes was analyzed by Kaplan–Meier (K-M) curve. Moreover, differences of clinicopathological features between different molecular subtypes in TCGA-LUAD cohort were compared.

Weighted gene co-expression network analysis

The key module genes linked to efferocytosis and molecular subtypes in LUAD were identified by the “WGCNA” R package (Zhang et al., 2022). Briefly, the enrichment score of ERGs in TCGA-LUAD cohort was computed by single-sample GSEA (ssGSEA). To ensure a scale-free network, the optimal soft threshold ( ) was determined by the pickSoftThreshold function. Next, average-linkage hierarchical clustering was conducted based on topological overlap matrix (TOM). Co-expressed gene modules conforming to the criteria of height = 0.25, deepSplit = 2, and minModuleSize = 10 were obtained. Further, the ssGSEA score and molecular subtypes of LUAD were employed as traits to delineate module-trait relationships. Critical module genes with the highest correlation coefficient were selected for following analysis (Wang et al., 2024).

DEGs and functional enrichment analysis

The DEGs with p.adj < 0.05 and |log2fold change (FC)| > log2 (0.5) between different LUAD subtypes were recognized employing the “limma” R package (Song et al., 2023). Then, the ERGs were screened by intersecting the DEGs with the module genes. Subsequently, specific functions of these candidate genes were investigated applying the “clusterProfiler” R package to perform Gene Ontology (GO) analysis in biological process (BP) term and Kyoto Encyclopedia of Genes and Genomes (KEGG) enrichment analysis (Zhao et al., 2021).

Development and verification of a RiskScore model

Firstly, univariate Cox regression analysis (p < 0.01) was performed on the candidate ERGs to select significant prognostic genes. The model was refined by LASSO regression analysis using the “glmnet” R package (Liu et al., 2023) and 5-fold cross-validation. Next, stepwise regression analysis was conducted to develop a prognostic gene signature linked to efferocytosis for LUAD. The prognostic RiskScore model based on TCGA-LUAD cohort was formulated to follow Duan et al. (2022): RiskScore= ∑βi ∗ExPi.

βi represents the coefficients of a gene in Cox regression model, ExPi is the gene expression.

Subsequently, the RiskScore was standardized by zscore and LUAD patients were separated into high- and low-risk groups by RiskScore = 0. Receiver operating characteristic (ROC) analysis was utilize to test the prediction accuracy of the RiskScore using the “timeROC” R package (Li et al., 2024). The OS rate in different risk groups was predicted applying K-M curve analysis. The robustness of the RiskScore model was validated based on GSE31210 dataset.

Immune cell infiltration analysis

To elucidate the association between the RiskScore and immune microenvironment, the “ESTIMATE” R package was employed to compute ESTIMATEScore, StromalScore, and ImmuneScore for different risk groups (Zhang et al., 2023a). Moreover, TIMER database (Liu et al., 2021) was applied to assess the infiltration of six immune cell types in all the risk groups, while immune microenvironment was characterized by MCP-counter algorithm (Chen et al., 2022).

Drug sensitivity analysis

The association between drug sensitivity and the RiskScore was analyzed based on the IC50 of chemotherapy drugs for each sample in TCGA-LUAD cohort utilizing the “oncoPredict” R package (Maeser, Gruener & Huang, 2021). Further, the correlation between the RiskScore, prognostic signatures and IC50 was analyzed, the drugs with false discovery rate (FDR) < 0.05 and |cor| > 0.4 were screened.

Single-cell atlas of LUAD

Cell types involved in LUAD progression were based on the scRNA-seq data of LUAD samples in GSE149655 dataset utilizing the “Seurat” R package (Zulibiya et al., 2023). Cells were clustered by the FindNeighbors and FindClusters functions. Afterwards, the top30 PCs were subjected to UMAP analysis for dimensionality reduction. Finally, the cell types of LUAD were annotated according to the marker genes from Cellmarker 2.0 database.

Cell cultivation and transfection

Human normal lung epithelial cell line BEAS-2B (C5382, BDBio, Hangzhou, China) and LUAD cell lines A549 (C6247, BDBio, China) and H2228 (C5122, BDBio, China) were all ordered from the Hangzhou BDBio Co., Ltd (Hangzhou, China) BEAS-2B cells were cultured in DMEM (C5382-500, BDBio, China) with 10% fetal bovine serum (FBS). A549 cells were cultivated in DMEM (C6247-500, BDBio, China) with 10% FBS, 1% penicillin/streptomycin (P/S) and one µg/mL puromycin. H2228 cells were cultured in RPMI-1640 (C5122-500, BDBio, China) with 10% FBS. All the cell lines were cultured in 5% CO2 at 37 °C. For silencing the expression of SEMA7A in LUAD cell lines, the siRNA of SEMA7A (si-SEMA7A: 5′-AGGCTTACGATGACAAGATCTAC-3′) and negative control (si-NC) were ordered from Shanghai GenePharma Technology Co., Ltd. Then, A549 and H2228 cells were transfected with siRNA applying Lipofectamine 2000 reagent (11668500, Invitrogen, Carlsbad, CA, USA).

Reverse transcriptional qPCR

The total RNA of BEAS-2B, A549 and H2228 cells isolated by Trizol reagent (R0016, Beyotime, Shanghai, China) was then reverse-transcribed into cDNA by the use of BeyoRT™ II cDNA Synthesis Kit (D7168M, Beyotime, China). Subsequently, RT-qPCR was conducted employing BeyoFast™ SYBR Green qPCR Mix (D7260-25ml, Beyotime, China) in ProFlex™ 96-well PCR System (4484075, Thermo Fisher Scientific, Waltham, MA, USA) (Zhang et al., 2023b). The primer sequences for amplification were shown in Table 1, with GAPDH as the housekeeping gene. The relative mRNA expressions of genes were calculated using 2−ΔΔCT method (Livak & Schmittgen, 2001).

Table 1 Primers for RT-qPCR amplification.

Gene	Primers (5′-3′)	
CD200R1	Forward: CGTCTCCCATTTGACTGGCAAC Reverse: CCAAATGAATCCCACGATGGTCA	
BTN2A2	Forward: CCGCTGTTACTTCCAAGAAGGC Reverse: CCAGATGCTCCCATCCTCTTGG	
DNASE2B	Forward: GCCAGCTCATCAGAGATTCCTG Reverse: TGAGCCATCCAGGCTGCAAAGA	
STAP1	Forward: GGAGGATTGAGACAGAGCAGAG Reverse: CTTCTGGAGCATCTCAGTTGCC	
SAMD9	Forward: GGGAACTACCTTGGCTATGCAC Reverse: CGTATTCCTGACGGTTCATTGCC	
BIRC3	Forward: GCTTTTGCTGTGATGGTGGACTC Reverse: CTTGACGGATGAACTCCTGTCC	
SEMA7A	Forward: CTTCTTCCGAGAGGACAATCCTG Reverse: GTGTTCCACTTGGAGACTGACAG	
GAPDH	Forward: GTCTCCTCTGACTTCAACAGCG Reverse: ACCACCCTGTTGCTGTAGCCAA	

CCK-8 assay

The transfected A549 and H2228 cells were seeded into 96-well plate at the concentration of 2 × 104 cells/well. At 24, 48, and 72 h, 10 µL CCK-8 solution (AC10873, Acmec Biochemical, Shanghai, China) was filled into each well for 2-h incubation. Finally, the absorbance value at 450 nm was determined under a microplate reader (GM3000, Promega, Madison, WI, USA).

Wound-healing test

Wound-healing test was conducted to detect the migratory ability of LUAD cells. The A549 and H2228 cells transfected with si-NC or si-SEMA7A were seeded into a 6-well plate (1 × 105 cells/well). After reaching confluence, a sterile scraper was employed to wound the cells, which were then washed with phosphate buffer and further cultivated in fresh DMEM at 37 °C for 48 h. Finally, the representative images at 0 h and 48 h were taken employing a DM IL LED inverted microscopy (Leica, Wetzlar, Germany), and the wound closure rate (%) was calculated by Image J (version 1.8.0) software.

Transwell test

Transwell test was utilized to assess the invasion of the LUAD cells. Transwell chambers with polycarbonate membrane (pore size: eight µm, Corning, NY, USA) were pre-coated with diluted Matrigel. The transfected H2228 and A549 cells (1 ×105 cells/well) were filled into the upper chamber with 200 µL non-serum medium, whereas the lower chamber was supplemented with complete medium as an attractant. After incubation for 48 h, the cells in lower chamber were fixed by 4% paraformaldehyde (P0099-100ml, Beyotime, Shanghai, China) and then dyed by 0.1% crystal violet (C0121-100ml, Beyotime, China). Finally, the invaded cells were observed under the previously used microscope (Fan, 2023).

Statistical analysis

All statistical analyses were conducted in the R software (version 4.0.1) and GraphPad Prism7.0. Log-rank test was utilized to compare prognostic differences between different groups. Spearman method was applied for correlation analysis. Significant difference between different groups was compared by t-test or ANOVA methods. Each experiment was independently repeated in triplicate, and the results were shown by means ± standard deviation (SD). The condition of statistical significance was set as p < 0.05.

Results

Two molecular subtypes of LUAD were identified using prognosis-related ERGs

Univariate Cox regression analysis revealed 29 ERGs significantly linked to the prognosis of LUAD, including 17 protective genes (hazard ratio (HR) < 1) and 12 risk genes (HR > 1) (Fig. 1A). Thereafter, consensus clustering analysis of these 29 prognostic ERGs classified two molecular subtypes (C1, C2) of LUAD at the consensus matrix k = 2 (Fig. 1B). PCA demonstrated clear separation between C1 and C2 subtypes (Fig. 1C), indicating the rationality of LUAD subtype classification. Furthermore, prognosis difference between two molecular subtypes was compared according to K-M curve. It was found that C2 subtype had a lower OS rate and poorer survival outcome than C1 subtype (Fig. 1D). In addition, in TCGA-LUAD cohort, compared to C1 subtype, C2 subtype had more male patients, and Stage III/IV, Pathologic_T2/T3, Pathologic_N2, Pathologic_M1, and ≤60 Age of LUAD patients (Fig. 1E).

Figure 1 Identifying the molecular subtypes of LUAD using efferocytosis-related genes (ERGs).

(A) Univariate Cox regression analysis screened prognosis-related ERGs in the TCGA-LUAD cohort; (B) Consensus clustering heatmap of LUAD patients; (C) PCA plot of two LUAD molecular subtypes; (D) Kaplan–Meier (K-M) curves were plotted to display the overall survival (OS) between C1 and C2 subtypes; (E) Clinical features distribution of C1 and C2 in the TCGA-LUAD cohort.

WGCNA identified 823 key module genes associated with efferocytosis in LUAD

The ssGSEA showed a lower enrichment score of ERGs in LUAD samples than that in control samples (Fig. 2A). The optimal soft-thresholding power (β = 12) was determined based on a scale-free topology criterion (correlation coefficient R2 > 0.85) (Fig. 2B). Subsequently, eight co-expressed gene modules were classified by average-linkage hierarchical clustering analysis (height = 0.25, deepSplit = 2, minModuleSize = 10) (Fig. 2C). Module-trait relationship heatmap displayed that the magenta module had strongest correlation with ssGSEA score (cor = 0.83) and LUAD subtypes (cor = 0.69) (Fig. 2D). Thus, 823 genes in the magenta module were extracted as critical efferocytosis-related module genes in LUAD for subsequent analysis.

Figure 2 Recognizing the critical module genes related to efferocytosis in LUAD by WGCNA.

(A) Enrichment score of ERGs in LUAD samples and control samples calculated by ssGSEA; ****p < 0.0001; (B) Screening of the optimal soft threshold power ( ) to develop a scale-free network; (C) Clustering dendrogram according to topological overlap matrix (TOM); (D) Module-trait relationships between ssGSEA score, LUAD subtypes and modules.

A total of 601 candidate ERGs in LUAD were mainly implicated in immune-relevant pathways

A sum of 1,593 DEGs between subtypes C1 and C2 were screened (p.adj < 0.05 and |log2FC| > log2) and visualized into a volcano plot (Fig. 3A). Then, 601 genes in the intersection between the 1,593 DEGs and 823 module genes were considered as the candidate ERGs in LUAD (Fig. 3B). Hereafter, KEGG enrichment analysis revealed that cytokine-cytokine receptor interaction, cell adhesion molecules (CAMs), chemokine signaling pathway, etc. were the primary pathways enriched by the 601 candidate genes (Fig. 3C). GO-BP enrichment analysis demonstrated that these genes were principally implicated in the regulation of lymphocyte activation and leukocyte activation, leukocyte differentiation, T cell activation, etc. (Fig. 3D). Collectively, these results suggested that the 601 candidate ERGs might be closely implicated in the inflammation and immune-relevant pathways.

Figure 3 Screening efferocytosis-related candidate genes of LUAD and functional enrichment analyses.

(A) DEGs between C1 and C2 subtypes were visualized into volcano plot; (B) Venn diagram of DEGs and module genes for screening candidate genes; (C) The candidate genes were subjected to KEGG pathway enrichment; (D) GO-BP enrichment analysis on the candidate genes.

A 7-ERG RiskScore model was established and exhibited high robustness in the prognostic evaluation of LUAD

Univariate Cox regression analysis identified 68 significant prognostic genes from the 601 candidate genes. To refine the RiskScore model, LASSO regression analysis showed that the model was the optimal at lambda = 0.0235 (Figs. 4A–4B). Afterwards, seven prognostic ERGs in LUAD were identified by stepwise regression analysis, including four protective genes (CD200R1, BTN2A2, STAP1, DNASE2B) and three risk genes (SAMD9, SEMA7A, BIRC3) (Fig. 4C). Then, the RiskScore was formulated as: “RiskScore = −0.49*CD200R1−0.32*BTN2A2−0.174*DNASE2B−0.223*STAP1+0.149*SAMD9+ 0.272*BIRC3+0.165*SEMA7A”.

Figure 4 Development and verification of the prognostic RiskScore for LUAD.

(A) Coefficient of each variable in LASSO regression analysis; (B) Confidence intervals under lambda; (C) Prognostic signature genes and their coefficients; (D) Risk type, survival status, and signature gene expression in the TCGA-LUAD cohort; (E) ROC curve was generated for the RiskScore and K-M curve of OS in different risk groups in the TCGA-LUAD cohort; (F) Survival status, risk type, and expressions of the signature genes in the GSE31210 dataset; (G) ROC curve was plotted for the RiskScore and K-M curve of OS in different risk groups in the GSE31210 dataset.

Further, based on the cutoff of “RiskScore = 0”, 217 patients were categorized into high-risk group, whereas 273 patients were categorized into low-risk group in the TCGA-LUAD cohort (Fig. 4D). Compared to low-risk group, CD200R1, BTN2A2, STAP1, and DNASE2B were low-expressed, while SAMD9, SEMA7A, and BIRC3 were high-expressed in high-risk group (Fig. 4D). The 1-, 2-, 3-, 4-, and 5-year AUC of the RiskScore reached 0.76, 0.71, 0.73, 0.74, and 0.71, respectively (Fig. 4E), indicating strong prediction performance of the model. K-M curve demonstrated a lower OS rate and worse outcome in high-risk group than those in low-risk group (Fig. 4E). Additionally, the robustness of the RiskScore model was confirmed in the validation set (GSE31210), showing similar results as the training set (Figs. 4F–4G).

Correlation of immune microenvironment, drug sensitivity and the RiskScore

ESTIMATE analysis revealed that ESTIMATEScore, StromalScore, and ImmuneScore were lower in high-risk group (Fig. 5A). TIMER and MCP-counter algorithms also demonstrated that high-risk LUAD patients had notably lower infiltration of most immune cells such as macrophages, T cells, DCs, cytotoxic lymphocytes, B cells, CD4 T cells, CD8 T cells (Figs. 5B–5C). These findings showed an immunosuppressive microenvironment in the high-risk group, which may be closely linked to the cancer progression and a worse prognosis of the high-risk LUAD patients. Furthermore, eight chemotherapy drugs, namely, Axitinib_1021, AZD8055_1059, BMS.754807_2171, Doramapimod_1042, GSK269962A_1192, JQ1_2172, PRIMA.1MET_1131, and Ribociclib_1632, were discovered to be markedly correlated with the RiskScore in the TCGA-LUAD cohort (FDR < 0.05 and |cor| > 0.4) (Fig. 5D). In particular, Doramapimod_1042 was closely linked to the RiskScore (R = 0.65, p < 2.2e−16) (Fig. 5E). The IC50 of Doramapimod_1042 in high-risk group was notably higher in comparison to low-risk group (Fig. 5F), suggesting that LUAD patients with a lower RiskScore might be more sensitive to the drug. These results may facilitate the drug selection process in personalized management of LUAD.

Figure 5 Correlation between the RiskScore, immune microenvironment, and drug sensitivity.

(A) ESTIMATEScore, StromalScore, ImmuneScore in different risk groups; (B–C) TIMER and MCP-counter assessing the infiltration scores of 6 and 10 immune cells in different risk groups; (D) Association of RiskScore, prognostic signatures and drug IC50 in the TCGA-LUAD cohort; (E) Relationship between Doramapimod_1042 IC50 and RiskScore; (F) IC50 value of Doramapimod_1042 in different risk groups; ****(p < 0.0001); ***(p < 0.001); **(p < 0.01); *(p < 0.05).

The single-cell atlas of LUAD was delineated by scRNA-seq analysis

The scRNA-seq analysis was performed on the GSE149655 dataset to delineate the single-cell atlas of LUAD. A sum of 4,054 cells were obtained after filtration, standardization, dimensionality reduction and then clustered into 12 subpopulations (Fig. 6A). These cells were further annotated by the marker genes into nine cell types (Fig. 6B), namely, epithelial cells (CLDN3, EPCAM), T cells (CD3E, CD3D), macrophages (ACP5, AIF1), mast cells (TPSAB1, MS4A2), endothelial cells (CLDN5, FCN3), fibroblast cells (DCN, LUM), DCs (LAMP3, MFSD2A), B cells (BANK1), and plasma cells (MZB1, JCHAIN) (Fig. 6C). The abundance of each cell type in different LUAD patients of GSE149655 dataset was shown in Fig. 6D. Additionally, the expressions of the seven prognosis signature genes in different cell types of LUAD were detected, and it was observed that these genes were mainly expressed in mast cells (Fig. 6E). This indicted that mast cells might function crucially in the development of LUAD.

Figure 6 Delineation of the single cell atlas of LUAD by scRNA-seq analysis.

(A–B) UMAP plots of cell clusters before and after annotation; (C) Marker genes in each cell type; (D) Abundance of each cell type in different LUAD samples of GSE149655 dataset; (E) Expression of seven prognostic signature genes in each cell type.

In vitro cell-based assays were conducted to validate the expressions and potential functions of the key genes

RT-qPCR revealed that the relative mRNA expressions of CD200R1, BTN2A2, SAMD9, BIRC3 and SEMA7A were all overexpressed in A549, H2228 cells (two LUAD cell lines) than BEAS-2B cells (human normal lung epithelial cell line) (Fig. 7A). Since the expression of SEMA7A was most significantly elevated in LUAD cells and has been observed to be closely linked to drug resistance and tumor metastasis in a variety of cancers (Yang et al., 2024), it was selected as a representative gene for subsequent functional validation experiments. CCK-8 assay displayed that after silencing SEMA7A, the cell viability of A549 and H2228 cells was all notably reduced (Figs. 7B–7C). Furthermore, wound healing and Transwell assays demonstrated that SEMA7A knockdown markedly lowered the rate of wound closure (Fig. 7D) and the number of invaded cells (Fig. 7E) of A549 cells. Similar results were observed in H2228 cells (Figs. 7F–7G). Collectively, silencing SEMA7A markedly suppressed the abilities of LUAD cells to proliferate, migrate and invade.

Figure 7 In vitro assays for validating the role of prognostic signature gene in LUAD cells.

(A) RT-qPCR detected the relative mRNA expressions of seven prognosis genes in BEAS-2B (human normal lung epithelial cell line) and A549 and H2228 (two LUAD cell lines); (B–C) Cell viability of A549 cells (B) and H2228 cells (C) transfected with si-NC or si-SEMA7A was detected by CCK-8 assay; (D–E) Wound healing assay (D) and Transwell assay (E) detected the effects of SEMA7A silencing on the migration and invasion of A549 cells; (F–G) Wound healing assay (F) and Transwell assay (G) detected the impacts of SEMA7A silencing on the H2228 cell migration and invasion. All procedures were independently conducted in triplicate (n = 3). ****(p < 0.0001); ***(p < 0.001); **(p < 0.01); *(p < 0.05); ns (no significant difference).

Discussion

Clinical treatment outcomes of LUAD remain unsatisfactory, despite recent advancements in screening and therapeutic methods (Chen, Wang & Hu, 2021). The identification of robust prognostic signatures may improve the survival of LUAD patients. Efferocytosis, a crucial process to clear apoptotic cells and maintain homeostasis (Xu et al., 2023), functions critically in the progression of cancers including LUAD (Liu & Wei, 2023). However, studies that systematically explored efferocytosis-related prognostic predictors for LUAD are limited. Based on the ERGs, our current study classified two molecular subtypes of LUAD with different survival outcomes and clinical features. Hereafter, the seven prognostic ERGs in LUAD were identified to establish a RiskScore model to evaluate the prognosis, immune infiltration, and drug sensitivity in LUAD. Overall, our findings may facilitate the prognosis evaluation and personalized treatment of LUAD.

The RiskScore model for LUAD was established based on four protective ERGs (CD200R1, BTN2A2, STAP1, DNASE2B) and three risk ERGs (SAMD9, SEMA7A, BIRC3), showing high robustness in the prognostic prediction for LUAD. CD200R1 is a member of immunosuppressive receptors principally expressed in immune cells such as CD4+/CD8+ T cells and myeloid cells, while its ligand CD200 is present in various normal and tumor cells (Fenaux et al., 2023). The CD200:CD200R1 interaction could modulate the TME, accelerating tumor growth and metastasis (Chang et al., 2020; Bisgin et al., 2019). Unfavorable survival in non-small cell lung cancer (NSCLC) may be related to high-expressed CD200R1, whereas CD200R1 knockdown could inhibit the cancer cell proliferation (Yoshimura et al., 2020). In LUAD, CD200R1 has been identified as one of the immune checkpoints-related genes correlated with patients’ clinical stages and prognosis (Hua et al., 2023). BTN2A2, an immunoregulatory gene of the butyrophilin family, has inhibitory effects on T cell activation and is implicated in immune tolerance in cancers (Brunschwiler et al., 2024; Bowler et al., 2018). The frameshift mutation of BTN2A2 might facilitate the pathogenesis of colon cancer via modulating immune response (Kim et al., 2023). The expression level of BTN2A2 is relevant to the clinical outcomes, cell proliferation and migration in breast cancer (Chen et al., 2024). BTN2A2 is also considered as a potential biomarker and treatment target for glioma (Wang et al., 2023). STAP1 is a member of the signal-transducing adaptor protein family that functions importantly in immune response and tumor occurrence. STAP1 in microglia could promote the progression of glioma (Yang et al., 2023a). The methylation of STAP1 in peripheral blood immune cells shows potential diagnostic and prognostic values for treating hepatocellular carcinoma within five cm (Sun et al., 2024a). DNASE2B, a deoxyribonuclease, is expressed in sialaden, lung and few other tissues without clear function, though its involvement in lens fiber cell differentiation has been manifested in rats (Ishida et al., 2020; Mori et al., 2022). SAMD9 is a sterile α motif domain-containing protein that locates at the chromosome 7q21.2. Previous study reported an association between SAMD9 and several malignancies (Buonocore et al., 2017). For instance, SAMD9 overexpression restrains tumor genesis and progression of NSCLC (Ma et al., 2014). The miRNA/SAMD9 signaling has important effects on modulating cisplatin chemoresistance in NSCLC (Wu et al., 2016). SEMA7A, alternatively known as CD108, is an immune semaphorin that regulates a variety of immunoinflammatory processes such as cytokine secretion, inflammatory infiltration and immune cell interplay (Yang et al., 2023b). It has been reported that SEMA7A contributes to tumor metastasis via promoting extracellular matrix adhesion or alluring epithelial-mesenchymal transition in oral cancer, melanoma and breast cancer (Liu et al., 2018; Ma et al., 2015; Black et al., 2016). SEMA7A could enhance EGFR-TKI resistance in LUAD cells with EGFR mutation (Kinehara et al., 2018). Our current study observed that silencing SEMA7A notably inhibited the proliferation, invasive and migratory capabilities of LUAD cells. BIRC3 is an apoptosis inhibitor and its dysregulation affects the inflammatory and immune responses in tumorigenesis, progression and therapy resistance (Zhang et al., 2024b). Taken together, we speculated that these genes were crucially involved in LUAD progression and can serve as the potential therapeutic targets and prognostic genes for LUAD.

Furthermore, analysis on the correlation between immune infiltration and the RiskScore demonstrated that compared to the low-risk LUAD group, the infiltration scores of most immune cells such as B cells, Macrophages, T cells, CD4 T cells, CD8 T cells, DCs and Cytotoxic lymphocytes were lower in high-risk group. Previous study suggested that efferocytosis strongly influences the immune cell activities in TME and facilitates a tolerogenic microenvironment (Tajbakhsh et al., 2021). Besides, efferocytosis facilitates tumor development by forming an immunosuppressive TME via eliminating dead cells and producing anti-inflammatory cytokines (Chen, Li & Li, 2024). These results demonstrated that the immunosuppressive microenvironment in high-risk LUAD group were closely linked to the progression of LUAD and poor prognostic outcomes. In addition, efferocytosis-related molecules are involved in the chemotherapy resistance of tumors (Cheng et al., 2022). We identified eight drugs strongly correlated with the RiskScore, and low-risk LUAD patients were more sensitive to Doramapimod_1042. Doramapimod is a potent anti-inflammatory compound for the treatment of rheumatoid arthritis, human endotoxemia, and Crohn’s disease (Bauquier et al., 2020; Branger et al., 2003; Schreiber et al., 2006). Overall, these outcomes could provide reference for the risk assessment and drug selection in personalized management of LUAD.

Nonetheless, our present study had several limitations. Firstly, although the RiskScore model was validated using public databases in this study, the single-cell data source was relatively limited and lacked prospective clinical sample validation, which may affect the generalization and clinical applicability of the model. Further study will incorporate multicenter and prospective clinical samples, with a particular focus on evaluating the expressions and prognosis significance of the key model genes in independent cohorts to enhance clinical translation potential. Secondly, the drug sensitivity analysis based on oncoPredict in this study relied on computational biology prediction results, which should be validated by cellular and animal experiments. Subsequent in vitro and in vivo experiments will be carried out to verify differential drug responses between low- and high-risk groups, thereby further assessing the model’s clinical utility for guiding personalized treatment. Finally, the limited number of LUAD samples in our scRNA-seq dataset may not fully reflect the tumor heterogeneity and immune microenvironment complexity. To address this, we plan to integrate newly generated single-cell transcriptomic data containing LUAD patients across different stages, treatment histories, and outcomes to improve our understanding of cellular heterogeneity and to enhance the reliability of the current findings.

Conclusion

To conclude, the current research classified two molecular subtypes of LUAD with distinct survival outcomes and clinical characteristics, contributing to the risk assessment of patients with LUAD. A RiskScore model was established based on seven ERGs, exhibiting strong performance in the evaluation of the immune status, drug sensitivity and prognosis for LUAD patients. These findings could help optimize the treatment strategies of LUAD, improve patients’ clinical outcomes, and promote the development of precision medicine in LUAD.

Supplemental Information

Supplemental Information 1 The scRNA-seq results after quality control in GSE149655 dataset

(A) The nFeature_RNA, nCount_RNA, and percent.mt of LUAD samples; (B) UMAP plot of LUAD samples.

Supplemental Information 2 167 ERGs used in this study

Supplemental Information 3 MIQE checklist

Abbreviations

ANOVA analysis of variance

AUC area under ROC curve

BP biological process

CAMs cell adhesion molecules

CCK-8 Cell Counting Kit-8

DC dendritic cell

DEGs differentially expressed genes

DMEM Dulbecco’s modified eagle medium

ERGs efferocytosis-related genes

FBS fetal bovine serum

FC fold change

FDR false discovery rate

GEO Gene Expression Omnibus

GO Gene Ontology

HR hazard ratio

IC50 half-maximal inhibitory concentration

KEGG Kyoto Encyclopedia of Genes and Genomes

K-M Kaplan–Meier

LASSO least absolute shrinkage and selection operator

LUAD lung adenocarcinoma

MCP-counter microenvironment cell populations-counter

OS overall survival

PCA principal component analysis

P/S penicillin/streptomycin

RNA-seq RNA sequencing

ROC receiver operating characteristic

RT-qPCR Reverse transcriptional quantitative real-time PCR

scRNA-seq single-cell RNA-seq

SD standard deviation

si small interfering

ssGSEA single sample gene set enrichment analysis

TCGA the Cancer Genome Atlas

TIMER tumor immune estimation resource

TME tumor microenvironment

TOM topological overlap matrix

TPM transcript per million

UMAP uniform manifold approximation and projection

WGCNA weighted gene co-expression network analysis

Additional Information and Declarations

Competing Interests

Author Contributions

Data Availability

The authors declare there are no competing interests.

Zengmin Dai conceived and designed the experiments, performed the experiments, analyzed the data, prepared figures and/or tables, authored or reviewed drafts of the article, and approved the final draft.

Shaofeng Jin performed the experiments, analyzed the data, authored or reviewed drafts of the article, and approved the final draft.

Shanshan Huang performed the experiments, analyzed the data, authored or reviewed drafts of the article, and approved the final draft.

Bingyang Liu performed the experiments, analyzed the data, authored or reviewed drafts of the article, and approved the final draft.

Xingkai Shen performed the experiments, analyzed the data, authored or reviewed drafts of the article, and approved the final draft.

Yuhong Jin conceived and designed the experiments, analyzed the data, prepared figures and/or tables, authored or reviewed drafts of the article, and approved the final draft.

The following information was supplied regarding data availability:

Sequences are available at NCBI: GSE31210, GSE149655

All raw data is available at GitHub at Zenodo:

- https://github.com/ZengminDai/all-raw-data.git.

- ZengminDai. (2025). ZengminDai/all-raw-data: all raw data (v.1.1.1). Zenodo. https://doi.org/10.5281/zenodo.15123031.

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
