# Peer review of "Development of a prognostic RiskScore model using efferocytosis-related signature genes for lung adenocarcinoma"

_PeerJ, doi:10.7717/peerj.19892_

## Round 0.1 · original submission · Major Revisions

While Reviewer 1 suggested minor revisions, the substantive concerns raised by Reviewer 2 require thorough consideration. After a comprehensive evaluation, I believe major revisions are necessary before your manuscript can be considered for publication. Please address all reviewers' comments in detail and submit a point-by-point response along with your revised manuscript.

Reviewer 1 ·

Basic reporting

no comment

Experimental design

1. Why was SEMA7A chosen for the follow-up experiments? If only SEMA7A is validated, then this study would be focusing on the impact of SEMA7A on lung adenocarcinoma, making the title inaccurate.
2. The abstract needs to be more concise. It is suggested to separate the sections of methods and results.
3. What is the sample size in Fig. 7? The sample size needs to be supplemented.
4. It is recommended to use literature from the past three years as much as possible. Please make the necessary revisions.
5. In the phrase “using pickSoftThreshold function, the optimal soft threshold ( ) was determined. N,” the text within the parentheses is missing.
6. For the formula, please use a solid line (/) instead of a horizontal line, and variables should be italicized.
7. All fonts in the figures should be uniformly set to Times New Roman.
8. All names in the figures must be consistent with those in the main text. Please check.

Validity of the findings

no comment

Reviewer 2 ·

Basic reporting

In this study, the sample information of LUAD was downloaded from a public database. The molecular subtypes of LUAD were classified according to ERGs, and the prognosis and clinical characteristics were analyzed. A RiskScore model with prognostic signature genes was developed and validated, and in vitro experiments were conducted for further verification. This study provides new ideas for prognostic assessment.

Experimental design

no comment

Validity of the findings

no comment

Additional comments

1. In the introduction section, the description of the currently commonly used treatment methods for LUAD is not detailed enough. It is advisable to point out the corresponding latest targeted therapies and immunotherapies, as well as the role of this study in these treatments, so as to highlight the significance of this research.
2. Efferocytosis is an important research direction of this study. Therefore, its definition should be described in detail in the introduction section.
3. Briefly describe the specific situation of the ERGs, instead of merely stating that they were acquired from a relevant literature.
4. In the cell experiment method, the 2-ΔΔCt method needs to cite relevant literature.
5. The overall structure of the article conforms to the PeerJ standard, but the language needs further polishing. Some sentences are long and have many grammatical errors.
6. In the wound healing test section, it mentions fresh DMEM or RPMI-1640 medium. Please specifically describe which type of cell is cultured in which medium to avoid confusing the readers.
7. There are many literatures on similar prognostic models in LUAD, and there is an overlap of related genes. It is recommended to compare and describe the model of this study with the previously obtained models.
8. The results show that the molecular subtypes C1 and C2 are enriched in different pathways. Relevant discussions are necessary.
9. This study also compared the drug sensitivities of the high-risk and low-risk groups. It is necessary to further discuss the related drugs involved.
10. The description in the limitations section is too general. The descriptions of relevant animal experiments and in vivo experiments in the future should be more specific.
11. The introduction part covers the relationship between efferocytosis and tumor progression, but it is necessary to supplement the literature in recent years (such as related studies in 2023-2024) to highlight the timeness of the research.
12. Figure 5F should clearly mark the threshold of statistically significant difference (e.g. p < 0.05). The description of QC results in supplementary Figure S1 is also required.

---

## Round 0.2 · accepted · Accept

Both reviewers have recommended acceptance of your revised manuscript. Based on their positive feedback, I am pleased to inform you that your paper has been accepted for publication.

Reviewer 1 ·

Basic reporting

no comment

Experimental design

no comment

Validity of the findings

no comment

Additional comments

By integrating phagocytosis related genes and multi-omics analysis, a prognostic model of LUAD with clinical translation potential was constructed in this study, which not only deeps the understanding of tumor microenvironment, but also provides new targets for precision treatment.

Reviewer 2 ·

Basic reporting

In this study, LUAD sample data were retrieved from publicly available databases. Molecular subtypes of LUAD were systematically classified based on ERGs (endoplasmic reticulum genes), followed by comprehensive analysis of prognostic outcomes and clinical characteristics. A RiskScore prognostic model incorporating signature genes was constructed and validated, with in vitro functional experiments further performed to validate the model’s robustness. Collectively, these findings offer novel insights for LUAD prognostic assessment. Notably, the revised manuscript has demonstrated significant improvements in methodological rigor, data presentation, and interpretative depth; no additional comments are required at this stage.

Experimental design

no comment

Validity of the findings

no comment